# Intermittent light enhances pigment production in the diatom *Phaeodactylum tricornutum*: a combined physiological and transcriptomic approach

Jiahui Zheng,[1,2,3] Jumao Yuan,[4] Xinwei Wang,[5] Dahai Zhang,[6] Lin Lin,[4] Nuo Shi,[4] Yuanyuan Feng[1,2,3,7]

**ABSTRACT**  Diatoms are ecologically important microorganisms that rely on dynamic pigment regulation for light use and photoprotection. These pigments also possess antioxidant, UV-protective, and anti-inflammatory properties, making them valuable for both ecological and biotechnological applications. While pigment biosynthesis in diatoms is well studied, the molecular responses to intermittent light environments remain poorly understood. Here, we investigated the adaptation of the model diatom *Phaeodactylum tricornutum* to simulated seasonal and intermittent light–dark (LD) cycles using a combination of physiological and transcriptomic analyses. We assessed growth, pigment content, photosynthetic performance, and elemental composition under different LD regimens (LD 24:0, LD 16:8, LD 12:12, LD 8:16, LD 4:2, LD 2:2, and LD 2:4). Our results showed that intermittent light stimulated growth and photosynthetic efficiency, with the highest pigment content observed under the LD 2:2 regimen. This regimen also increased the ratio of photosynthetic to photoprotective pigments and reduced diadinoxanthin de-epoxidation. Transcriptomic data revealed that moderate LD cycles (LD 2:2 and LD 12:12) upregulated genes involved in chlorophyll and carotenoid biosynthesis, as well as light-harvesting complex pathways. This study provides key insights into the molecular and physiological strategies diatoms employ to adapt to variable light environments and highlights their potential for improved pigment production in industrial and biotechnological applications.

**IMPORTANCE**  Diatoms are tiny algae that play a key role in Earth's ecosystems by absorbing carbon and producing oxygen. They also create valuable pigments with properties that can protect against environmental stress and may have uses in health, food, and industry. This study reveals how diatoms adapt to changing light conditions, which are common in natural waters. Researchers found that short intervals of light and darkness can boost diatom growth and increase pigment production, especially under very short light cycles. By exploring the genes and biochemical processes involved, the study offers new insight into how these microorganisms survive in dynamic environments. This knowledge could help improve the sustainable production of diatom-derived pigments for a range of practical applications, from food coloring to natural health products.

**KEYWORDS**  diatoms, photosynthetic accessory pigments, light-dark cycle, fucoxanthin, intermittent light

Diatoms are one of the most important and abundant phytoplankton functional groups on the earth, contributing approximately 40% of global marine primary productivity (1). They contain a rich array of pigments, including chlorophyll *a*, chlorophyll *c*, beta-carotene, fucoxanthin, and others (2). Given the important roles these

**Peer Reviewer** Xin Lin, Xiamen University, Xiamen, Fujian, China

Address correspondence to Yuanyuan Feng, yuanyuan.feng@sjtu.edu.cn.

The authors declare no conflict of interest.

See the funding table on p. 18.

pigments play in light absorption and photoprotection, research into phytoplankton pigments is important to understanding diatom photosynthetic physiology (3). Diatoms enhance their light capture capabilities and dissipate excess light energy by adjusting the size of light-harvesting pigment antennae, the number of photosynthetic units, and the types of pigment-protein complexes, thereby adapting to varying light environments (4). Notably, the characteristic pigment of diatoms, fucoxanthin, pairs with chlorophyll a/c to form the light-harvesting protein fucoxanthin chlorophyll *a/c* protein (FCP) (5). Under low light conditions, the elevated level of light-trapping pigments in diatom cells helps meet the photosynthetic light requirements. Conversely, when diatom cells are exposed to excessive light, the xanthophyll cycle (XC) can dissipate excess excitation energy in the light-harvesting complex (LHC) of PSII as heat, thus providing photoprotection (6). The XC mainly comprises two processes: the diadinoxanthin (Dd) cycle and the violaxanthin (V) cycle. The former involves a de-epoxidation step to convert Dd into diatoxanthin (Dt), while the latter involves two de-epoxidation steps converting V to antheraxanthin (A) and subsequently to zeaxanthin (Z) (7).

The unique molecular structure of diatom pigments is believed to have special bioactivities, and the studies on the applications of diatom pigments across various industries are receiving much attention. For instance, Dt has been demonstrated to inhibit the expression of inflammatory factors interleukin-1β (IL-1β) and matrix metalloproteinase-9 (MMP-9), significantly reduce ultraviolet-induced reactive oxygen species (ROS) levels, and decrease extracellular nitric oxide (NO) release (8). Additionally, fucoxanthin has been shown to inhibit β-glucuronidase activity in DLD-1 cancer cells, thereby potentially slowing the invasion and metastasis of colon cancer (9). It can also improve insulin resistance by promoting the expression of β3-adrenergic receptor (Adrb3) and glucose transporter protein 4 (GLUT4) mRNA, which can mitigate symptoms of hyperglycemia, hyperinsulinemia, and hyperleptinemia (10). Violaxanthin has been recognized for its strong antioxidant and anti-inflammatory effects, significantly inhibiting nitric oxide (NO) and prostaglandin E2 (PGE2) in mouse macrophages, and ameliorating UVB-induced skin photoaging (11, 12). At present, diatom pigments have been commercially utilized in fields, such as obesity prevention, anti-inflammation, and anti-cancer, and are used as a feed additive in poultry and aquaculture. In recent years, the antioxidant, anti-wrinkle, anti-ultraviolet, antibacterial, anti-inflammatory, and anti-allergic attributes of diatom pigments are becoming more evident, highlighting their prospective significance in the cosmetics and skincare industries (13–18).

Pigment biosynthesis in diatoms is intricately associated with light conditions. Studies indicate that environmental factors, including light intensity, light quality, and ultraviolet radiation significantly affect the pigment content and composition in diatoms (19–22). In addition, another important environmental factor regulating the growth and photosynthesis of phytoplankton is the periodical fluctuation of light and dark conditions, including both diel and short-term variations (23). However, research on the effects of light-dark cycles on pigment synthesis in diatoms is relatively scarce. Preliminary studies suggest that the light-dark cycle may affect the total light absorption, cell division patterns, and organic matter synthesis in diatoms *Triceratium reticulum*, *Skeletonema* sp., and *Phaeodactylum tricornutum*, with these effects exhibiting species-specific traits and closely linked to the cell growth cycle (24). Moreover, freshwater diatoms, such as *Cyclotella meneghiniana* and *Stephanodiscus binderanus*, display distinct lipid metabolism patterns under varying light-dark cycles, and these metabolic shifts are closely tied to different stages of cell growth (25). Yu et al. (26) found that the duration of light exposure and the frequency of light-dark transitions significantly altered the chlorophyll *a* content in *Chlorella* sp., with the fastest-growing treatment displaying the lowest chlorophyll *a* levels; in contrast, the light-dark cycle had no significant effect on carotenoid content. Additionally, there is a notable increase in lipid yield in treatments subjected to alternating light-dark conditions (26). The effect of the light-dark cycle on diatom biosynthesis may be related to endogenous circadian rhythms, with the bHLH-PAS nuclear protein RITMO1 likely playing a key role in regulating these rhythms

in algal cells (27). Furthermore, studies have demonstrated that during dark treatment, the nuclear transcription activity in *P. tricornutum* significantly decreased. Despite a reduction in total pigment levels, the proportion of light-harvesting pigments remained stable, and the photosynthetic efficiency only slightly declined, with the photoprotection and photosynthetic mechanisms remaining functional to facilitate rapid recovery upon re-illumination (28).

In order to further understand the effects of photoperiods on different growth stages in marine diatoms and reveal the underlying molecular regulatory mechanisms, we carried out laboratory incubation experiments on diatom *P. tricornutum* under seven distinct light-dark cycles to simulate a range of seasonal and intermittent light-dark fluctuations. The growth rate, pigment synthesis, photosynthetic parameters, and cellular elemental content were examined. Furthermore, the transcriptomic data were analyzed to provide insights into the regulatory mechanisms of light-dark cycles on the physiological metabolism of diatoms.

## MATERIALS AND METHODS

### Algal strain and culture medium

The marine diatom *P. tricornutum* (CCMP 2561) strain was originally isolated from the coastal waters near Blackpool, England, North Atlantic (54°N, 4°W), obtained from the Bigelow National Center for Marine Algae and Microbiota (NCMA). Before the experiment, the stock culture was maintained in f/2 medium (29) at 20°C, with 12-h light:12-h dark under irradiance of 50 µmol photons·m$^{-2}$·s$^{-1}$.

### Experimental design and growth conditions

Seven experimental treatments were established, each with three biological replicates. The specific light regimens were set up as follows: (i) constant light (control): 24-h light:0-h dark (LD 24:0); (ii) long photoperiod: 16-h light:8-h dark (LD 16:8); (iii) medium photoperiod: 12-h light:12-h dark (LD 12:12); (iv) short photoperiod: 8-h light:16-h dark (LD 8:16); (v) intermittent long photoperiod: 4-h light:2-h dark (LD 4:2); (vi) intermittent medium photoperiod: 2-h light:2-h dark (LD 2:2); (vii) intermittent short photoperiod: 2-h light:4 h dark (LD 2:4).

The experiment was conducted in batch culture mode. Prior to the start of the batch incubation, the cultures were acclimated to the respective light regimens in 1-L incubation bottles for 14 days under semi-continuous incubation, with daily dilution using freshly made medium to maintain the cell abundance ~2 × 10$^6$ cells/mL. After acclimation, cultures in the exponential growth phase were inoculated into 2-L polycarbonate bottles (Nalgene, USA) at an initial cell density of 8 × 10$^4$ cells·mL$^{-1}$. The cultures were grown in autoclaved artificial seawater (30), supplemented with nutrients, trace metal, and vitamin solutions to f/2 level (29) at 20 °C with light intensity of 70 µmol photons·m$^{-2}$·s$^{-1}$. The light intensity used for the incubation was close to the optimal condition for fucoxanthin production of *P. tricornutum* (31, 32).

### Sampling and measurement methods

#### Cell growth

One milliliter of the culture was sampled, fixed with 6 µL Lugol's solution, and counted under a microscope using a 100 µL plankton counting chamber (33).

The growth curve of algae was fitted using the integral form of the logistic equation:

$$N = \frac{K}{1 + e^{(a - rt)}},$$

where $N$ represents the population abundance, $r$ is the intrinsic growth rate, $K$ is the carrying capacity of the environment, and $a$ is a constant parameter obtained after integration.

### Pigment contents

The samples for pigment analysis were collected by filtering 10 mL of culture through a GF/F membrane (Whatman, USA). The filters were then flash-frozen in liquid nitrogen and stored at −80°C until analysis. The frozen filter was cut into small pieces and placed in a centrifuge tube with 4 mL of N,N-dimethylformamide (DMF). The sample was sonicated in an ice-water bath (JY92-IIN, JINGXIN, 40 W, 1 min), centrifuged at 3,000 rpm for 3 min, and the supernatant was filtered through a 0.22-μm nylon filter. A 1-mL aliquot of the filtrate was transferred to a vial and mixed with 200 μL of Milli-Q water (34).

Pigment analysis was performed using high-performance liquid chromatography (HPLC, Shimadzu, Japan). Mobile phase A consisted of methanol/acetonitrile/pyridine aqueous solution (50:25:25, v/v/v), and mobile phase B consisted of methanol/acetonitrile/acetone (20:60:20, v/v/v). The column used was a ZORBAX Eclipse XDB-C8 (4.6 × 150 mm, 5 μm) with a Gemini C18 guard column (4 × 3.0 mm). The column temperature was maintained at 25°C with a flow rate of 1 mL/min. The gradient elution was run over 47 min as follows: 0–2 min, A: 95%, B: 5%; 2–22 min, A: 95%→60%, B: 5%→40%; 22–24 min, A: 60%, B: 40%; 24–30 min, A: 60%→5%, B: 40%→95%; 30–40 min, A: 5%, B: 95%; 40–42 min, A: 5%→95%, B: 95%→5%; 42–47 min, A: 95%, B: 5%.

### Elemental contents

Samples for particulate organic carbon (POC) and particulate organic nitrogen (PON) measurements were filtered onto GF/F membranes (pre-combusted at 450°C for 4 h). After being dried at 60°C, the filters were stored in desiccators and analyzed using an elemental analyzer (UNICUBE, Elementar, Germany).

Samples for particulate organic phosphorus (POP) measurements were filtered onto pre-combusted GF/F membranes, rinsed with 2 mL 0.17 M $Na_2SO_4$ solution, then transferred to pre-combusted glass bottles, dried in 2 mL 0.017 M $MgSO_4$ solution, and analyzed using the molybdenum blue method (35).

Biogenic silica (BSi) samples were filtered onto 0.6-μm polycarbonate membranes (Millipore, USA), dried, and analyzed spectrophotometrically (36). And for dissolved organic carbon (DOC) measurement, 30 mL of filtrate was collected after GF/F filtration, stored at −20°C, and measured using a TOC-L total organic carbon analyzer (Shimadzu, Japan).

### Photosynthetic parameters

Diatom cultures were placed in a temperature-controlled chamber and dark-adapted for 15 min. Photosynthetic parameters were then measured using a Phyto PAM Phytoplankton Analyzer (Walz, Germany). These parameters included the potential maximum photochemical efficiency of PSII ($F_v/F_m$), maximum relative electron transport rate ($rETR_{max}$, relative maximum electron transport rate), light saturation point ($I_k$), and efficiency of electron transport ($\alpha$).

## Data analysis

One-way analysis of variance (ANOVA) was performed using GraphPad Prism 7.0. The significance level was set at $P < 0.05$. Tukey's HSD test was used for multiple comparisons. Levene's test was conducted to assess homogeneity of variances. Significance was indicated using either asterisks or letters.

## Transcriptomic analysis

Culture cells (100 mL) from the late exponential growth phase at the light phase were filtered onto 0.6-µm polycarbonate membranes (Millipore, USA), flash-frozen in liquid nitrogen, and stored at −80℃. The samples were then sent to Novogene (Beijing, China) for transcriptome sequencing.

RNA was extracted and quality-assessed using the Agilent 2100 Bioanalyzer. High-quality RNA was enriched for mRNA using Oligo(dT) magnetic beads, followed by random fragmentation and cDNA synthesis. Library construction involved purification and selection of PCR products using AMPure XP beads, with final sequencing on the Illumina NovaSeq 6000 platform, generating 150 bp paired-end reads. Raw sequencing data were processed using CASAVA software (Illumina, USA) to produce fastq files. Quality control steps included removing adapter sequences, reads with N bases, and low-quality reads (Qphred ≤ 5 for more than 50% of the read length). Clean reads were aligned to the reference genome using HISAT2 (v2.0.5), with genome and annotation data sourced from public databases.

Gene expression was quantified using featureCounts (v1.5.0-p3), calculating reads mapped to each gene and normalizing for gene length and sequencing depth (FPKM). Differential expression analysis was conducted using DESeq2 (v1.20.0), applying a negative binomial distribution model. $P$-values were adjusted using the Benjamini-Hochberg method, with adjusted $P$-values ≤0.05 indicating significant differential expression. Functional enrichment analyzes, including GO and KEGG pathways, were performed using clusterProfiler (v3.8.1).

## RESULTS

### Growth conditions

The cell abundance of *P. tricornutum* under different light-dark cycles can be categorized into three distinct growth phases (Fig. 1a): the lag phase (Days 0–2), exponential phase (Days 3–6), and stationary phase (Days 7–10). The average cell abundance during the exponential phase in the different experimental treatments showed a general trend: long photoperiod > medium photoperiod > short photoperiod. Using the integral form of the logistic equation to fit the data, intrinsic growth rates were calculated for each treatment, as presented in Fig. 1b. Notably, the intrinsic growth rate of the 4:2 light-dark cycle treatment was significantly higher than those of the 2:4 and 16:8 treatments ($P < 0.05$).

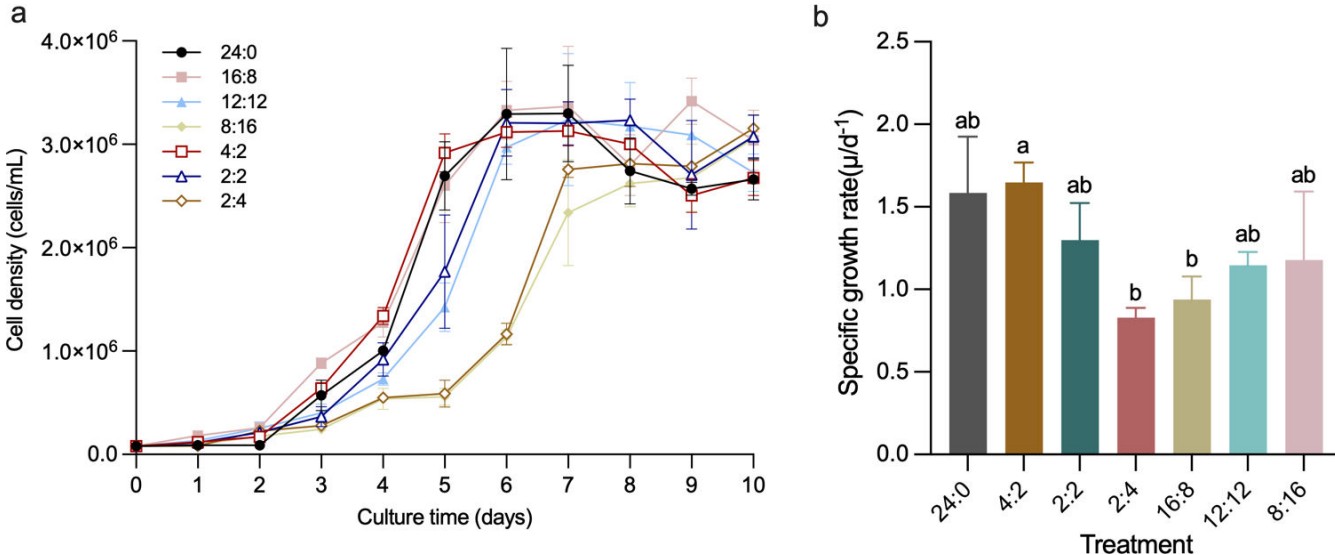

**FIG 1** (a) Changes in cell abundance of *Phaeodactylum tricornutum* under different light-dark cycles. (b) Comparison of intrinsic growth rates. Error bars represent standard deviation ($n = 3$).

## Pigment contents

### Pigment concentration and composition during the course of the incubation

The effects of light-dark cycles on the total pigment accumulation in the *P. tricornutum* culture system are shown in Fig. 2a. At the end of the culture period, the total pigment accumulation in the 2:2 treatment was significantly higher than in the other treatments ($P < 0.05$), reaching 2,025.26 ± 144.30 µg·L$^{-1}$.

The changes in pigment composition over time in control are shown in Fig. 2b. During the early exponential phase (Day 4), the proportion of chlorophyll *a* reached its peak at 46%, while during the other growth stages, it remained stable at approximately 39%. The proportion of diadinoxanthin reached 2.9%, about twice as high as in other stages. Meanwhile, the proportion of fucoxanthin was relatively low in the early exponential phase at 39.21%, stabilizing at around 42% during the other stages. In addition, the high proportion of chlorophyllide a during the lag phase (Day 2) at 18% reflects its accumulation as an intermediate, while its decrease on Day 4 (11.31 %) indicates its conversion into chlorophyll *a*.

For the long photoperiod treatments (Fig. 2c and d), across all growth stages, the total pigment content in the 16:8 treatment was lower than that in the 4:2 treatment. During the early exponential phase (Day 4), the proportion of chlorophyll *a* in the 4:2 treatment peaked at 50%, while the proportions of fucoxanthin and chlorophyllide a dropped to their lowest values (37% and 10%, respectively). These trends are similar to those observed in control. In contrast, these changes in the 16:8 treatment occurred earlier, during the lag phase (Day 2). Notably, the proportion of fucoxanthin in the 16:8 treatment showed a general upward trend, reaching its highest value at the end of the culture period, and the proportion of diadinoxanthin in the 16:8 treatment was higher than that in the 4:2 treatment throughout the experiment.

The changes in pigment composition over time in the medium photoperiod treatment are shown in Fig. 2e and f. The 2:2 treatment exhibited a large increase in pigment content, reaching its highest value during the stationary phase. In the early exponential phase (Day 4), the proportion of chlorophyll *a* in the 2:2 treatment reached its maximum, while the proportions of fucoxanthin and chlorophyllide a were at their lowest. In both the 12:12 and 2:2 treatments, chlorophyll *a* and chlorophyllide a displayed an inverse relationship throughout the experiment. The continuous light treatment showed a higher final proportion of fucoxanthin, a trend similar to that observed in the long photoperiod treatment.

The changes in pigment composition over time in the short photoperiod treatment are depicted in Fig. 2g and h. No significant difference was observed in total pigment content between the 2:4 and 8:16 treatments. In both treatments, the proportion of fucoxanthin reached its maximum by the end of the culture period. The proportion of chlorophyllide a increased daily, while the proportion of chlorophyll *a* showed a general decreasing trend.

Additionally, as shown in Fig. 2i and j, the proportion of violaxanthin and diatoxanthin in the culture system remained consistently low, accounting for less than 1% of the total pigment content. The results indicated that the proportion of diatoxanthin showed an overall increasing trend, while the content of violaxanthin exhibited a decreasing trend.

### Cellular pigment content

During the exponential phase (Fig. 3a), the fucoxanthin content in the 4:2 treatment was 0.18 ± 0.00 pg·cell$^{-1}$ compared with 0.14 ± 0.01 pg·cell$^{-1}$ in the 16:8 treatment, indicating that intermittent light significantly enhanced cellular fucoxanthin levels in *P. tricornutum* ($P < 0.05$). In the stationary phase (Fig. 3b), intermittent light also significantly increased cellular chlorophyll *a* content ($P < 0.001$), with the 4:2 treatment recording 0.16 ± 0.02 pg·cell$^{-1}$ compared with 0.1167 ± 0.01 pg·cell$^{-1}$ in the 16:8 treatment. Additionally, cellular violaxanthin content in the 4:2 treatment was significantly higher than in the 16:8 treatment ($P < 0.05$).

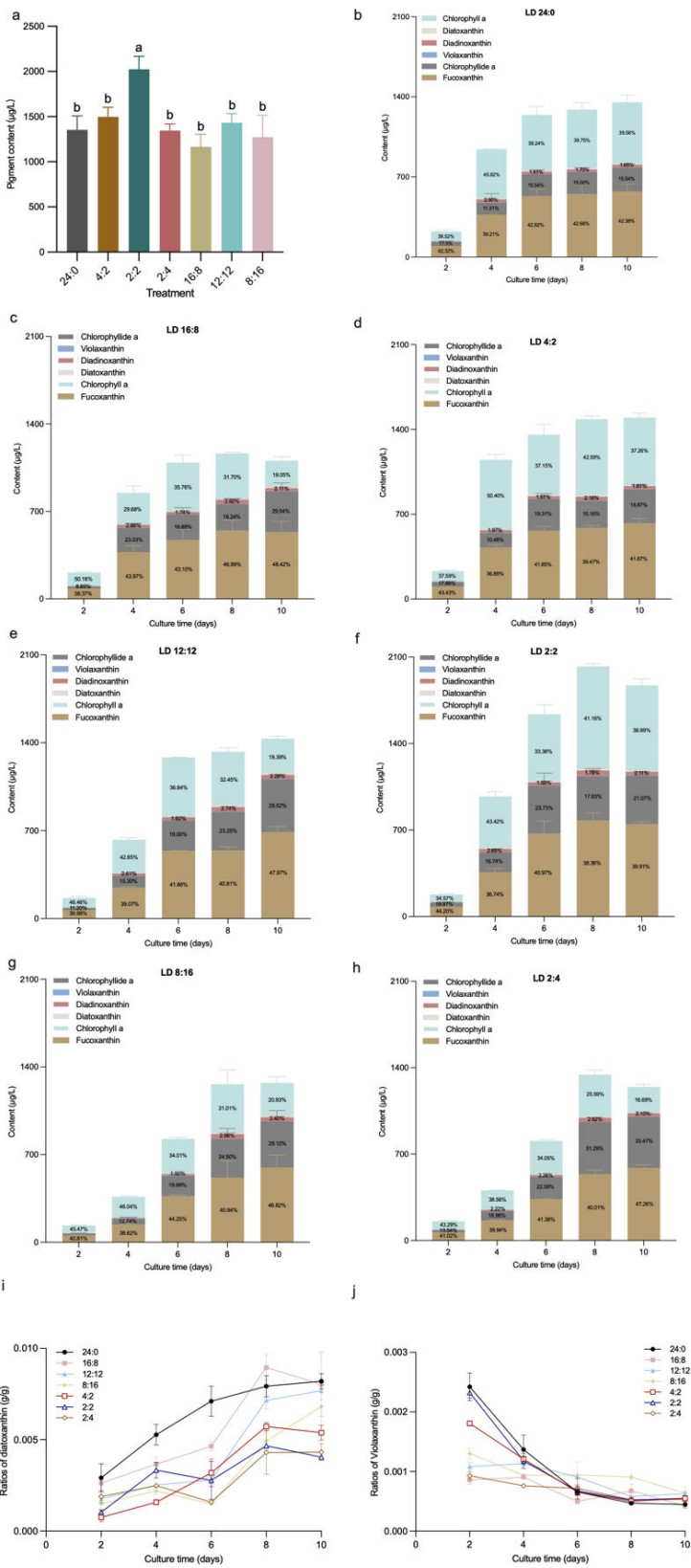

FIG 2 (a) Total pigment contents at the end of the cultivation period in all the experimental treatments. (b–h) Trends in pigment composition over time under different light-dark cycles. Variations in the concentration ratios of (i) diatoxanthin to total pigments and (j) violaxanthin to total pigments across the cultivation period; Error bars represent standard deviation ($n = 3$).

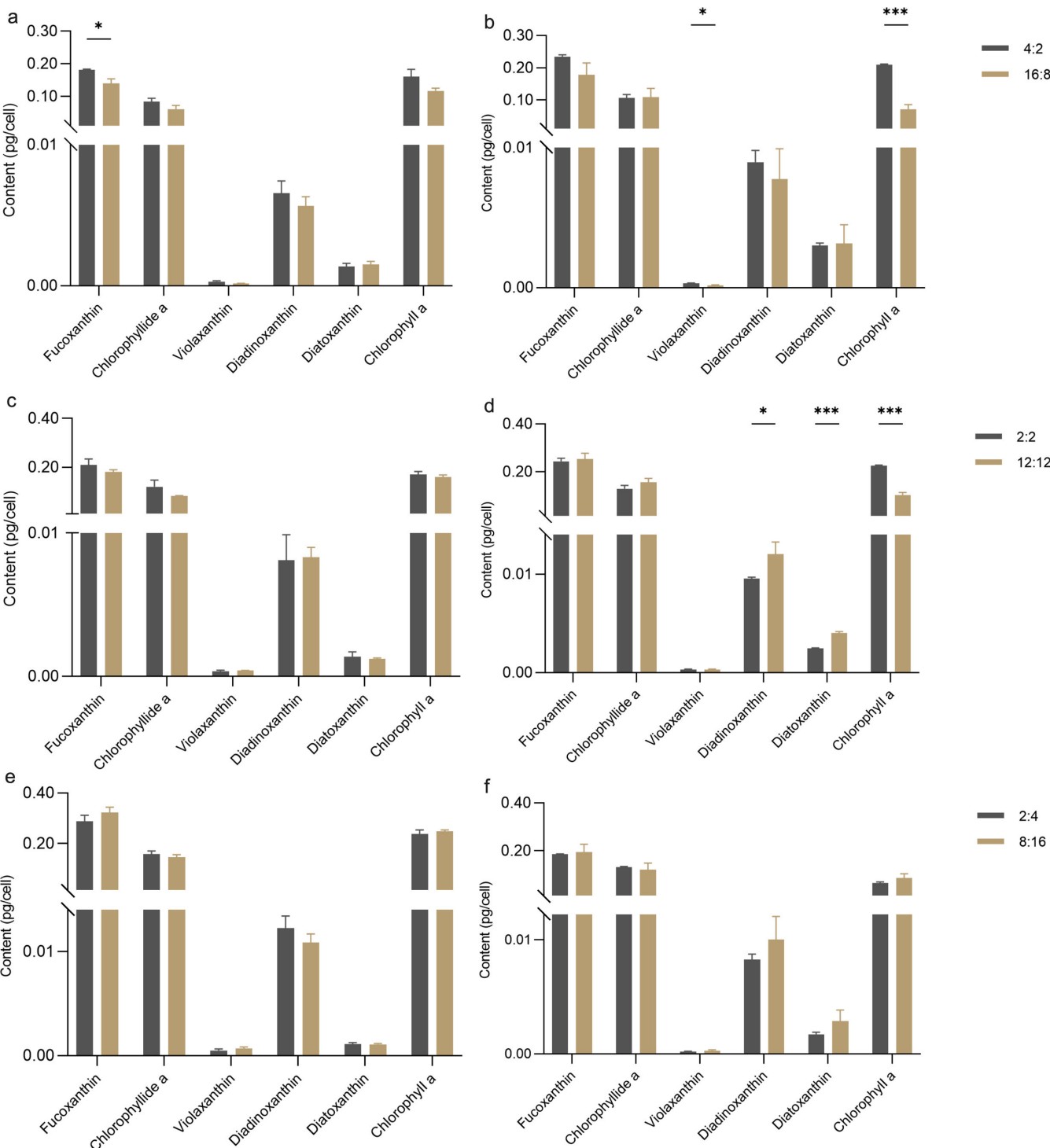

**FIG 3** Comparison of intracellular pigment content. (a) Long photoperiod at the exponential phase; (b) long photoperiod at the stationary phase; (c) medium photoperiod at the exponential phase; (d) medium photoperiod at the stationary phase; (e) short photoperiod at the exponential phase; (f) short photoperiod at the stationary phase. Data from day six represent the exponential phase, and day 10 represents the stationary phase. Error bars indicate standard deviation ($n$ = 3). ***$P$ < 0.001, *$P$ < 0.05.

In the medium photoperiod treatments during the exponential phase (Fig. 3c), no significant differences in cellular pigment content were observed between the 2:2 and 12:12 treatments during the exponential growth phase. However, in the stationary phase

(Fig. 3d), intermittent light significantly decreased cellular diadinoxanthin content ($P < 0.05$). Similarly, the cellular diatoxanthin content in the intermittent light treatment was significantly lower than in the continuous light treatment ($P < 0.001$). Intermittent light also significantly increased chlorophyll *a* content in the stationary phase ($P < 0.001$), with the 2:2 treatment showing $0.2251 \pm 0.0001$ pg·cell$^{-1}$ compared with $0.10 \pm 0.00$ pg·cell$^{-1}$ in the 12:12 treatment.

There were no significant differences in cellular pigment content observed between the two short photoperiod treatments, under both exponential growth phase and stationary growth phase (Fig. 3e and f).

### Cellular pigment ratios during the stationary phase

The activation state of the diadinoxanthin cycle is represented by the de-epoxidation state (DEPS), calculated as the ratio of diatoxanthin (Dt) to the sum of diatoxanthin and diadinoxanthin (Dd), expressed as DEPS = Dt/(Dt + Dd). Furthermore, the ratio of photosynthetic to photoprotective pigments can be determined using the formula (Chl *a* + Chlide + FX)/(Dd + Dt + V), where Chl *a* represents chlorophyll *a*, Chlide represents chlorophyllide a, FX represents fucoxanthin, and V represents violaxanthin. The Chl *a*/FX ratio indicates the relative abundance of pigments involved in photosynthesis, while the FX/POC and Chl *a*/POC ratios represent the relationship between photosynthetic pigment content and particulate organic carbon (POC) production.

During the stationary phase, the Chla/FX ratio in the 4:2 treatment was significantly higher than that in the 16:8 treatment ($P < 0.0001$, Fig. 4d). Similarly, the Chl *a*/POC ($P < 0.001$) and (Chl *a*+Chlide + FX)/(Dd+Dt + V) ratios ($P < 0.01$) were also significantly higher in the 4:2 treatment. In the 2:2 treatment, the Chl *a*/FX, (Chl *a*+Chlide + FX)/(Dd+Dt + V), and Chl *a*/POC ratios were significantly higher than those in the 12:12 treatment ($P < 0.0001$), while the DEPS ($P < 0.01$) and FX/POC ($P < 0.05$) ratios were lower in the intermittent light treatment than in the continuous light treatment. In the 2:4 treatment, the Chl *a*/FX, FX/POC, and Chl *a*/POC ratios were all lower than in the 8:16 treatment, while the photosynthetic to photoprotective pigment ratio was higher ($P < 0.05$).

### Photosynthetic parameters

During the early exponential growth phase, the $F_v/F_m$ values of the three experimental treatments with longer total light exposure durations (24:0, 4:2, 16:8) were significantly lower than those of the other treatments ($P < 0.05$, Fig. 5a). In the stationary phase, the $F_v/F_m$ value of the control remained significantly lower than those of the other experimental treatments ($P < 0.05$), and the value of continuous long-light (16:8)

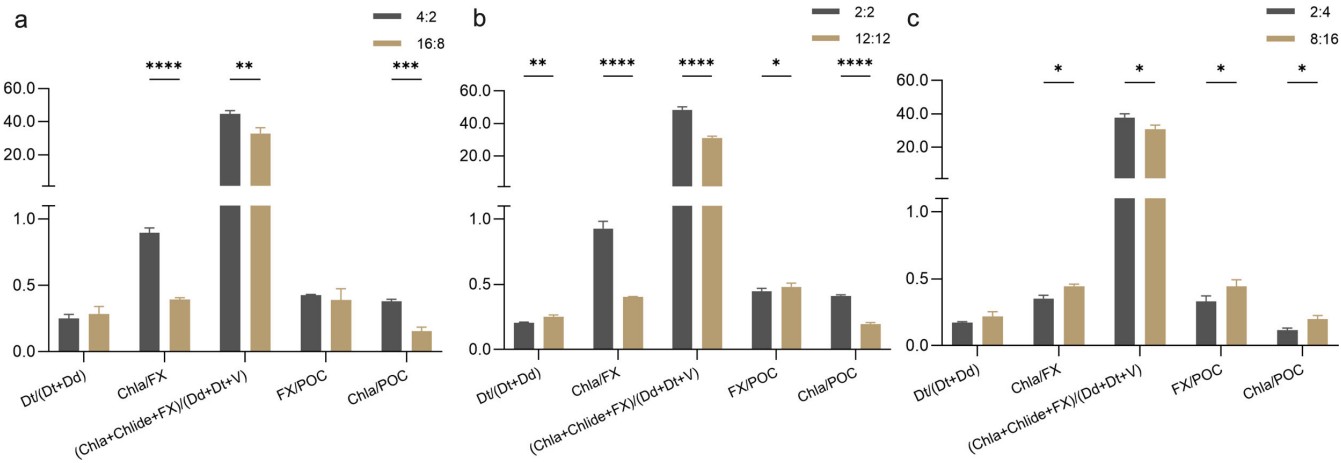

**FIG 4** Comparison of cellular pigment ratios during the stationary phase. (a) Long photoperiod (LD 4:2 vs. 16:8); (b) medium photoperiod (LD 2:2 vs. 12:12); (c) short photoperiod (LD 2:4 vs. 8:16). Error bars represent the standard deviation ($n = 3$), with statistical significance denoted as ****$P < 0.0001$, ***$P < 0.001$, **$P < 0.01$, and *$P < 0.05$.

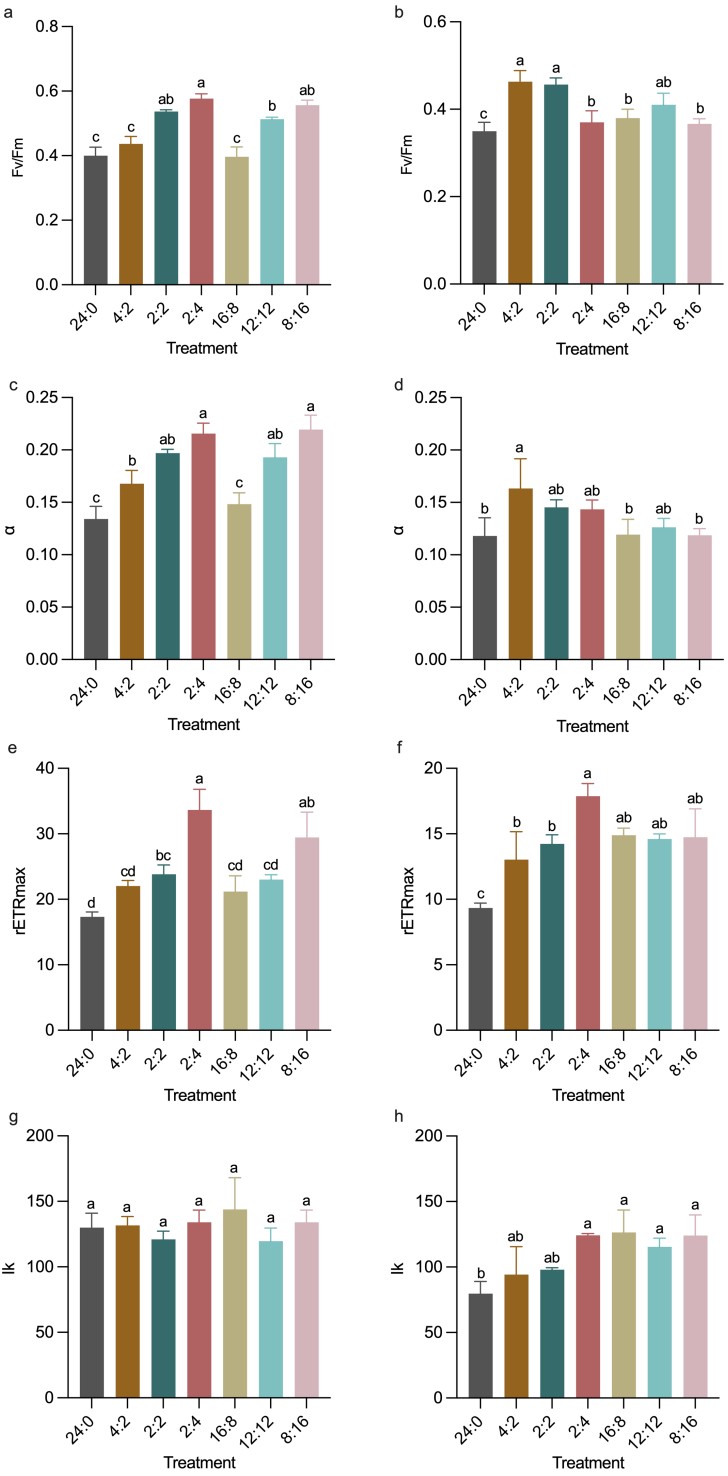

**FIG 5** The photo-physiological parameters: (a) $F_v/F_m$ at the exponential phase; (b) $F_v/F_m$ at the stationary phase; (c) α at the exponential phase; (d) α at the stationary phase; (e) $rETR_{max}$ at the exponential phase; (f) $rETR_{max}$ at the stationary phase; (g) $I_k$ at the exponential phase and (h) $I_k$ at the stationary phase under different light-dark cycles. Data from day 6 represent the exponential phase, and data from day 10 represent the stationary phase. Error bars represent standard deviations ($n = 3$). Different letters indicate significant differences between treatments ($P < 0.05$).

treatment was also significantly lower than the intermittent long-light treatment (4:2) ($P < 0.01$, Fig. 5b).

The α values of the control and 16:8 treatments were significantly lower than those of the other experimental treatments. Notably, the α value of the 4:2 treatment was significantly higher than that of the 16:8 treatment ($P < 0.05$, Fig. 5c). In the stationary phase, the differences between treatments became smaller, but within the long-light treatments, the higher α value was still observed in the 4:2 treatment ($P < 0.05$, Fig. 5d).

$rETR_{max}$, representing the maximum photochemical conversion efficiency of PSII and PSI per unit area, showed that, during the early exponential growth phase, the $rETR_{max}$ value of the short-light treatment was significantly higher than those of all other treatments ($P < 0.05$, Fig. 5e). In the stationary phase, the 2:4 treatment within the intermittent light treatments exhibited significantly higher $rETR_{max}$ values than the 4:2 and 2:2 treatments, while the control treatment had significantly lower values than the other experimental treatments ($P < 0.05$, Fig. 5f).

$I_k$, which represents the light intensity at which the photosynthetic rate reaches saturation, showed that in the stationary phase, the lowest $I_k$ value was observed at control ($P < 0.05$, Fig. 5g and h).

## Cellular elemental content

As shown in the diagram (Fig. S1a and b), during the exponential phase, cellular POC content in the short photoperiod treatment was significantly higher than in the long and medium photoperiod treatments ($P < 0.05$). In the stationary phase, the control treatment exhibited the highest cellular POC content, reaching $0.6005 \pm 0.2313$ pg·cell$^{-1}$. Additionally, the POC content in the 2:4 treatment was significantly higher than in the 8:16 treatment ($P < 0.05$).

During the exponential phase, cellular PON content in the short photoperiod treatment was significantly higher than in the long and medium photoperiod treatments ($P < 0.05$), with little impact observed from different light regimens. In the stationary phase, the cellular PON content at control was the highest, and the value of the 2:4 treatment was higher than that at 8:16 ($P < 0.05$) (Fig. S1c and d).

The cellular POP content at the short photoperiod condition was significantly higher than in the long and medium photoperiod conditions during the exponential phase ($P < 0.05$). In the stationary phase, cellular POP content in the continuous light treatments of the long and medium photoperiod treatments was significantly higher than in the corresponding intermittent light treatments ($P < 0.05$), with the 12:12 treatment exhibiting the highest content (Fig. S1e and f).

As shown in Fig. S1g, during the exponential phase, cellular BSi content at the short photoperiod condition was significantly higher than in the long and medium photoperiod treatments ($P < 0.05$). In the stationary phase, cellular BSi content in the 4:2 treatment was significantly higher than in the 2:2, 12:12, and 8:16 treatments ($P < 0.05$) (Fig. S1h).

## Transcriptome sequencing

### Differential expression analysis

To compare the gene expression differences between the LD 24:0 and LD 12:12, LD 24:0 and LD 2:2, and LD 12:12 and LD 2:2 samples, differential expression analysis was conducted using DESeq2. The criteria for selecting differentially expressed genes were DESeq2 padj < 0.05 and $|\log_2 FoldChange| > 1.0$. The results indicate that, compared with LD 24:0, there were 871 upregulated genes and 831 downregulated genes at LD 2:2; compared with the LD 12:12 treatment, there were 1,207 genes upregulated and 1,101 genes downregulated at LD 2:2; and compared with the LD 24:0 treatment, there were 1,034 genes upregulated and 850 genes downregulated at LD 12:12. These findings reveal variations in the fold change of differentially expressed genes at different LD cycles, suggesting that their responses to the LD cycle differ both in type and magnitude (Table 1; Tables S1 and S2).

**TABLE 1** Statistical table of differentially expressed genes

| Compare | All | Up | Down | Threshold |
|---|---|---|---|---|
| LD 2:2 vs LD 12:12 | 2,308 | 1,207 | 1,101 | DESeq2 padj <0.05, \|log$_2$FoldChange\| > 1.0 |
| LD 2:2 vs LD 24:0 | 1,702 | 871 | 831 | DESeq2 padj <0.05, \|log$_2$FoldChange\| > 1.0 |
| LD 12:12 vs LD 24:0 | 1,884 | 1,034 | 850 | DESeq2 padj <0.05, \|log$_2$FoldChange\| > 1.0 |

The principal component analysis (PCA) results suggest no overlap between the LD 24:0 and LD 12:12, LD 24:0, and LD 2:2, or LD 12:12 and LD 2:2 treatments, indicating that the samples from these three treatments, exposed to different light cycles, exhibit significant differences (Fig. 6).

*Enrichment analysis*

The top 20 significantly enriched KEGG pathways were visualized using a dot plot (Fig. S2). By comparison between LD 2:2 and LD 12:12, the most significantly enriched pathways were carbon metabolism, biosynthesis of secondary metabolites, and glyoxylate and dicarboxylate metabolism. By comparison of LD 2:2 vs. LD 24:0, the enriched pathways primarily included ribosome, biosynthesis of secondary metabolites, and glycolysis/gluconeogenesis. For the LD 12:12 vs. LD 24:0 comparison, the most significantly enriched pathways were biosynthesis of secondary metabolites, glycolysis/gluconeogenesis, and biosynthesis of cofactors. These results suggest that changes in photoperiod may influence the physiological responses of *P. tricornutum* by regulating key biological processes, such as carbon metabolism, protein synthesis, and energy metabolism.

Notably, the "biosynthesis of secondary metabolites" pathway was significantly enriched across all comparisons, indicating that this pathway is broadly and strongly regulated under different light-dark conditions. This pathway includes modules related to the synthesis of various pigments, including carotenoids. Further analysis revealed

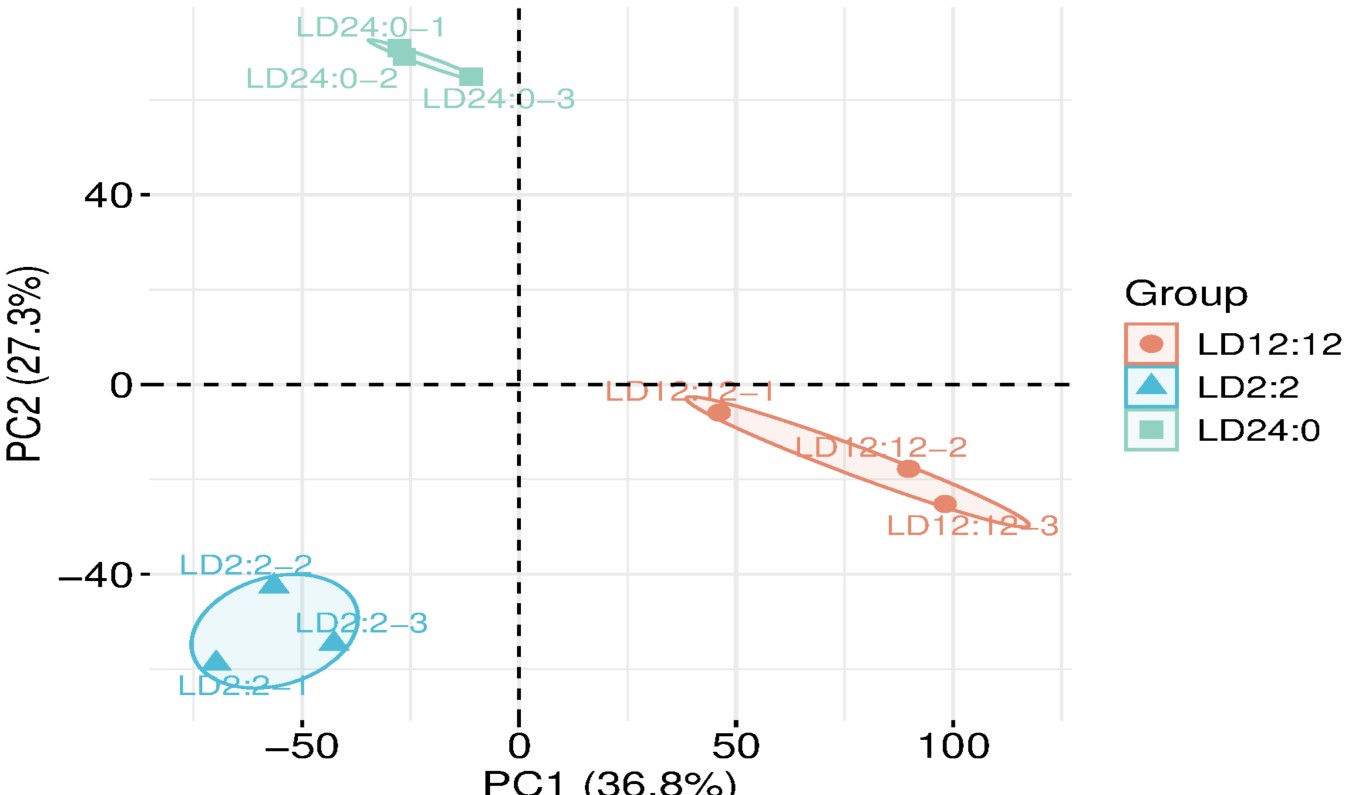

**FIG 6** Principal component analysis of different gene experiments among treatments.

that most key enzyme-encoding genes involved in carotenoid biosynthesis (e.g., PSY, PDS, CRTISO) exhibited significant differential expression among the different LD conditions. Given the essential role of pigment biosynthesis in maintaining photosystem stability and providing photoprotection, the prominent enrichment of this pathway suggests its central role in the adaptive response of *P. tricornutum* to photoperiod changes.

### Pigment synthesis pathway

In order to elucidate the regulatory mechanisms underlying light-driven pigment metabolism, we further investigated the expression patterns of key genes involved in the carotenoid and chlorophyll biosynthesis pathways. Here, a secondary screening of differential genes related to pigment synthesis was conducted, combining gene functional annotations. A heatmap was generated to further display the expression trends of these genes under different LD treatments (Fig. 7). Compared with the control, genes associated with carotenoid biosynthesis (PDS, PSY, ZEP1, ZEP3, VDL2, CRTISO1, CRTISO5) and chlorophyll biosynthesis (PHATRDRAFT_46085) were upregulated under moderate light conditions (LD 2:2 and LD 12:12). Additionally, a key carotenoid cleavage enzyme gene (PHATRDRAFT_37960) was significantly upregulated in control. Moreover, most genes related to light-harvesting proteins exhibited higher expression levels under these two light-dark cycle conditions. Overall, the heatmap results reveal significant transcriptional regulation differences in pigment synthesis and regulatory genes in *P. tricornutum* under different light cycle treatments.

In the carotenoid biosynthesis pathway, upstream enzymes, such as PSY, PDS (PDS-like1, PHATR_43904), ZDS (PHATR_43904), and CRTISO (CRTISO1/5), showed significant expression changes, indicating that light-dark rhythms strongly regulate pigment precursor synthesis. Downstream genes involved in the conversion of fucoxanthin and other photoprotective pigments (e.g., Ddx, Dtx), including VDE (VDL2) and ZEP (ZEP1/3), as well as carotenoid degradation genes such as CCD (PHATR-DRAFT_37960) and NCED (PHATRDRAFT_43374), also exhibited distinct and dynamic regulatory patterns.

In the chlorophyll biosynthesis pathway, key enzyme genes, such as PORC (PHATR-DRAFT_56588) and CHLG (PHATRDRAFT_46085), were also regulated by the light-dark cycle, reflecting the adaptive modulation of core photosynthetic pigments in response to varying light conditions. Additionally, differentially expressed genes associated with light-harvesting complexes were identified, including members of the Lhcf family (Lhcf2–5, Lhcf8–12, Lhcf16–17) and the Lhcr family (Lhcr1, Lhcr6–8, Lhcr10–11, Lhcr13). Notably, stress-responsive genes in the Lhcx family, such as Lhcx1 and Lhcx3, were upregulated under intermittent light conditions.

Overall, *P. tricornutum* modulates the expression of key genes in both carotenoid and chlorophyll biosynthesis pathways in response to different light-dark cycles, thereby maintaining efficient light energy conversion and enhancing its ability to cope with light-induced stress in complex light environments (Fig. 8).

## DISCUSSION

This study systematically examined the effects of light-dark cycles on the growth, pigment synthesis, photosynthetic performance, and cellular elemental composition of *P. tricornutum* under controlled light regimens. Our results suggest significant physiological responses to different light-dark cycles: moderate light durations promoted growth and enhanced photosynthetic efficiency, whereas the continuous light treatment (LD 24:0) induced declined photosynthetic efficiency and photoinhibition. In contrast, shorter light durations (LD 8:16, LD 2:4) restricted total light energy acquisition, resulting in reduced growth rates. Furthermore, the light-dark cycle significantly affected the intracellular contents and ratios of key pigments—such as chlorophyll *a*, fucoxanthin, and violaxanthin—at different growth stages, as well as the utilization of elements, such as carbon, nitrogen, and silicon. Overall, these findings suggest that the light-dark cycle

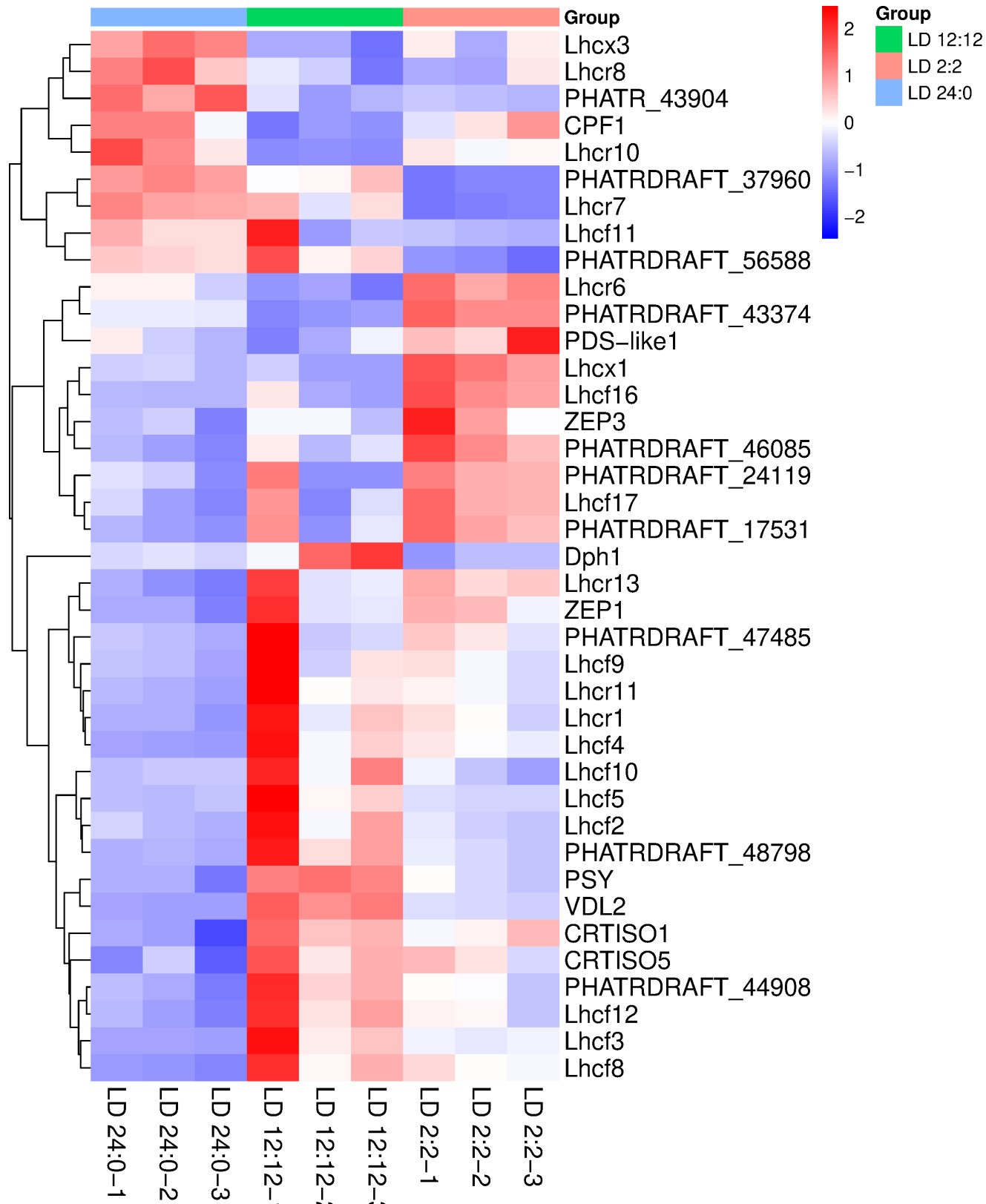

**FIG 7** Heatmap of differentially expressed genes related to pigment biosynthesis. (Based on the union of differentially expressed genes identified under LD 2:2, LD 12:12, and LD 24:0 treatments, a heatmap was generated using Z-score normalized FPKM values. Color scale bar indicates relative expression levels to the average values across treatments. Gene names are shown on the right, and treatments are indicated by the color bar at the top.)

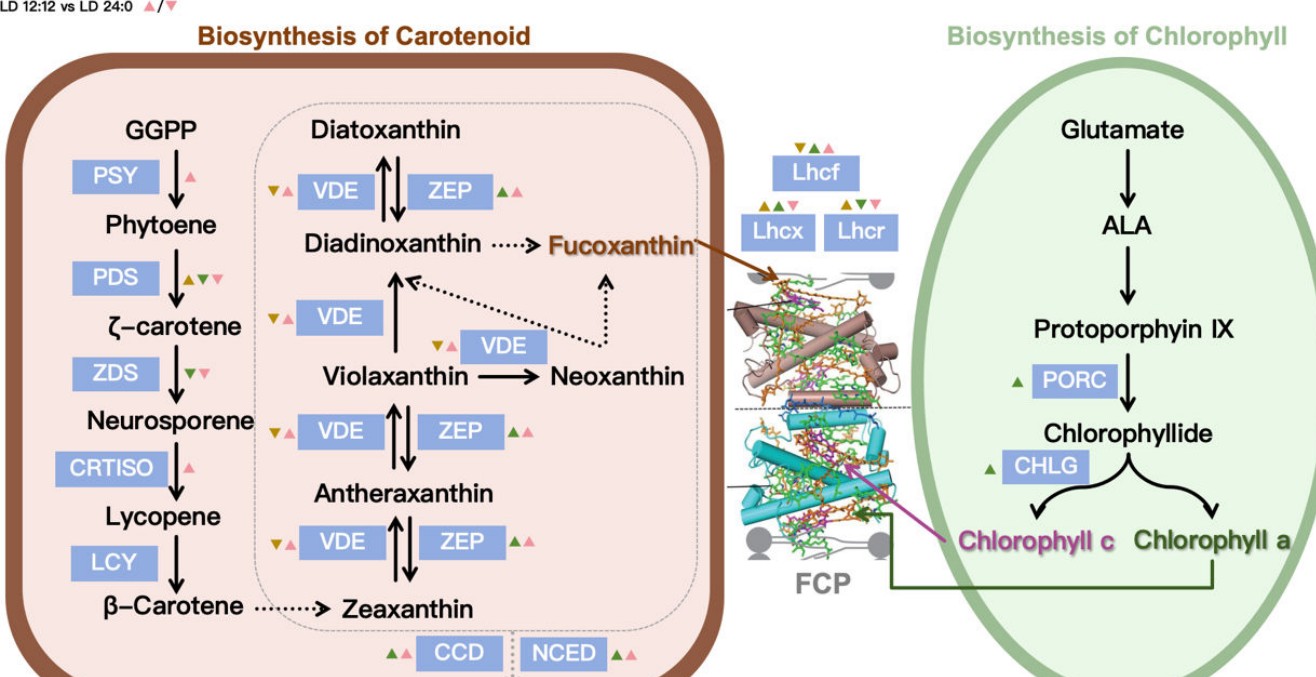

**FIG 8** Pigment biosynthesis pathway and differential gene expression under light/dark cycles in *P. tricornutum*. (Key genes in the pigment biosynthesis pathway are marked with blue boxes. Dashed lines indicate uncharacterized or putative biochemical reactions. The top-left legend compares three light/dark treatments, where upregulated and downregulated genes are represented by upward-pointing (▲) and downward-pointing (▼) triangles of different colors. FCP: fucoxanthin-chlorophyll protein complex in diatoms, composed of chlorophyll *a*, chlorophyll *c*, and fucoxanthin; GGPP: geranylgeranyl pyrophosphate synthase; PDS: phytoene desaturase; ZDS: ζ-carotene desaturase; CRTISO: carotenoid isomerase; LCY: lycopene cyclase; VDE: violaxanthin de-epoxidase; ZEP: zeaxanthin epoxidase; CCD: carotenoid cleavage dioxygenase; NCED: 9-cis-epoxycarotenoid dioxygenase; PORC: protochlorophyllide oxidoreductase C; CHLG: chlorophyll synthase.)

is a crucial environmental factor in regulating photosynthetic and metabolic processes in diatoms.

## Impact of light-dark cycles on pigment synthesis

The light-dark cycle affects both pigment synthesis and accumulation, as well as the regulation of the light-harvesting system. The physiological parameters examined reveal significant changes in the chlorophyll and carotenoid content of diatoms under different light cycles. Compared with the constant light control (LD 24:0) and the short photoperiod (LD 8:16, LD 2:4), moderate light durations (long and medium light treatments) promote pigment synthesis. Additionally, under the same light duration, intermittent light exposure is more beneficial for maintaining higher pigment levels compared to continuous light exposure.

The observed changes in pigment content may be related to the light adaptation strategy of diatoms: Extreme continuous light triggers negative feedback regulation of pigment synthesis and activates light protection mechanisms. In contrast, appropriate periods of darkness help restore and balance photosynthesis, preventing excessive light energy absorption that could cause light damage (37). Short light durations, however, are not favorable for photosynthetic productivity and pigment synthesis (38). A previous transcriptomic study confirmed the hypothesis regarding light damage. After 24 h of continuous light exposure, diatoms activated an intrinsic "biological clock inhibition" mechanism, downregulating the expression of genes involved in light-harvesting pigment synthesis to limit photosynthetic efficiency and avoid light damage,

while genes related to cell proliferation were significantly upregulated (39). Similarly, our transcriptomic sequencing results suggest that, compared with the continuous light treatments, moderate light durations significantly upregulated the expression of key enzyme genes involved in carotenoid synthesis, such as PSY, CRISO, and VDE, and also upregulated the expression of genes related to chlorophyll synthesis, such as PORC and CHLG. An earlier study pointed out that under the LD 12:12 light-dark cycle, the growth and division of diatom cells exhibit a clear periodic change, with pigment synthesis and cell growth occurring only during the light period and stopping during the dark period. Chlorophyll *a* content increases with cell growth and peaks just before division (40). Fluctuations in light intensity influence a range of cellular processes, including cell division, and diatoms adapt effectively to these changes by modulating the expression of key cell cycle regulators, such as cyclins and cyclin-dependent kinase genes (41, 42). These studies also help explain why pigment content in the short light duration treatments remained consistently low in the present study. It is possible that increased frequency of light-dark transitions may help balance extended photosynthetic activity (light capture) with photoprotection, thereby enhancing pigment synthesis efficiency (28).

## Responses of pigment components to light-dark cycles

Moreover, the experimental results showed that different types of pigments respond differently to LD cycles. For example, during the exponential phase, the intracellular fucoxanthin content was lower in the control and long light treatments, while it was higher in the medium and short light treatments. The overall trend for chlorophyll *a* content was short light > medium light > long light > control, similar to fucoxanthin. The content of diatoxanthin was particularly high in the LD 24:0 treatment, significantly higher than in other treatments ($P < 0.001$). Consistently, for the polar diatom *Fragilariopsis cylindrus*, under longer light durations, the content of the main light-harvesting pigment fucoxanthin was reduced, but not for the chlorophyll c and β-carotene contents (40). Similar light adaptation adjustments were observed for the phase changes of diatoxanthin and β-carotene during the light-dark cycle differing from those of chlorophyll (40). Conversely, under short light cycles, diatoms tend to maintain higher light-capturing pigment content to maximize the limited light exposure. Long light exposure also increased the pool capacity and NPQ ability of lutein cycle pigments (39).

These findings imply that when light durations are extended, the contents of protective pigments, such as diatoxanthin, are increased to dissipate excess energy, in order to avoid light damage from prolonged light exposure or high light intensity. Notably, as the culture progressed, an inverse trend was consistently observed between the levels of chlorophyllide *a* and chlorophyll *a*. In the chlorophyll *a* biosynthetic pathway described by Nymark et al. (43), the final step involves the conversion of chlorophyllide *a* into chlorophyll a, catalyzed by chlorophyll synthase. Therefore, the opposite trends observed may reflect the accumulation of chlorophyllide a as a precursor in chlorophyll *a* biosynthesis. The proportion of diatoxanthin in the total pigments increased over time, reflecting its accumulation as a light-protective pigment. In contrast, the proportion of violaxanthin decreased daily before stabilizing, which is the upstream pigment of diatoxanthin in the pigment synthesis pathway of *P. tricornutum* (7).

## Regulation of light-harvesting complexes by light-dark cycles

The light-harvesting complexes (LHCs) and related proteins were also regulated by the photoperiod. The primary light-harvesting antenna in diatoms is the fucoxanthin-chlorophyll binding protein (FCP) (44). Sequencing results suggest that, compared with control, the expression of genes related to the Lhcf family proteins (Lhcf 2–5, 8–12, 16) in FCP was significantly upregulated in the medium light treatment, while the expression of the blue light receptor gene CPF1 was downregulated. This suggests that genes encoding light-harvesting antenna proteins are also regulated by light-dark cycles. This

hypothesis is partially supported by the literature: circadian rhythm core genes identified in *Skeletonema costatum* and *P. tricornutum* include genes encoding photosynthetic antenna proteins, and their expression is regulated by light cycles (45).

Furthermore, Madhuri et al. knocked out the blue light receptor PtAureo1a in P. *tricornutum* and found that the mutation affected the circadian expression rhythm of a series of light reaction-related genes, including Lhcx1. In the wild type, Lhcx1 exhibited a regular pattern to adapt to diurnal light intensity variations, but in the mutant lacking the light receptor, this rhythm disappeared (46). In this experiment, differential expression of the Lhcx1 gene was also detected, with significantly higher expression levels in the LD 2:2 treatment compared with the LD 24:0 and LD 12:12 treatments. This suggests that, under high-frequency light-dark transitions, *P. tricornutum* adapts to changes in the external light environment by upregulating Lhcx1 gene expression, playing a protective role. The light-dark cycle not only affects pigment content changes but also alters the dynamic balance between light capture and light protection complexes through gene expression regulation. Additionally, the transcriptomic sequencing results revealed differential expression of photoreceptor pigments (Dph1), which also reflects the molecular mechanism by which diatoms sense and transmit light signals (47).

## Impact of light-dark cycles on photosynthetic efficiency and ecological implications

Photosynthetic efficiency and photochemical parameters were also regulated by the light-dark cycle. The photosynthetic parameters measured in this study varied with changing light cycles: shorter and moderate light-dark cycles help maintain higher photochemical efficiency, while excessively long or 24-h light exposure leads to a decline in photosynthetic parameters. Similarly, a previous study (39) suggests that when light duration was extended from 18 to 24 h, $F_v/F_m$ in *F. cylindrus* significantly decreased. In addition, *F. cylindrus* compensated for the negative effects of continuous light by increasing non-photochemical quenching, showing buffering capacity (39).

Additionally, under equal total light durations, the photosynthetic parameters of the exponential growth phase were always higher in the intermittent light conditions (LD 2:2, 4:2, and 2:4) than the continuous light treatments with the same light exposure time (LD 12:12, 16:8, and 8:16). It has been demonstrated that *P. tricornutum* indeed possesses an endogenous circadian rhythm: in continuous light, spontaneous fluorescence in wild-type cells still maintains periodic oscillations (27, 40). However, previous studies on diatom circadian rhythms mostly focused on conventional light-dark cycles like LD 24:0 (48, 49) and LD 12:12 (50), with limited understanding of the effects of intermittent light exposure. Our results reveal that higher frequency of light fluctuation may promote both photosynthetic performance and pigment synthesis efficiency. This suggests a potential acclimation strategy for diatoms in dynamic marine environments, such as turbulent waters with fast mixing or regions subject to periodic cloud cover.

## Conclusion

Overall, our results suggest that different light-dark cycles significantly affect the growth rate, pigment content, elemental composition, and photosynthetic parameters of *P. tricornutum*. Transcriptomic data under different light-dark cycle conditions reveal that light-dark cycles adapt to different light regimens by regulating the expression of light capture system and pigment metabolism-related genes. Based on the results of this study, future research could further investigate the interaction effects between light-dark cycles and light intensity, as well as the long-term adaptation mechanisms of diatoms.

## ACKNOWLEDGMENTS

The authors would like to thank Dr. Cong Zeng for constructive discussion on the transcriptomic analysis and Dr. Zhuoyi Zhu for helping with the pigment analysis, who are both affiliated with the School of Oceanography, Shanghai Jiao Tong University.

This study was financially supported by the National Natural Science Foundation of China (42276093), Deep Blue Foundation of Shanghai Jiao Tong University (SL2022PT203), and the Impact and Response of Antarctic Seas to Climate Change funded by the Chinese Arctic and Antarctic Administration (IRASCC 1-02-01B) to Y.F.; and National Key Research and Development Program of China (2023YFC3108400) to D.Z.

## AUTHOR AFFILIATIONS

[1]State Key Laboratory of Submarine Geoscience, School of Oceanography, Shanghai Jiao Tong University, Shanghai, China
[2]Key Laboratory of Polar Ecosystem and Climate Change, Ministry of Education, Shanghai, China
[3]Shanghai Key Laboratory of Polar Life and Environment Sciences, Shanghai, China
[4]Shanghai Kedai Biotechnology Co., LTD, Shanghai, China
[5]School of Marine Sciences, Ningbo University, Ningbo, Zhejiang, China
[6]Key Laboratory of Marine Chemistry Theory and Technology, Ministry of Education, Ocean University of China, Qingdao, Shandong, China
[7]Laboratory for Polar Science, Polar Research Institute of China, Ministry of Natural Resources, Shanghai, China

## AUTHOR ORCIDs

Jiahui Zheng  http://orcid.org/0009-0008-4474-5924
Dahai Zhang  http://orcid.org/0000-0001-7105-1583
Yuanyuan Feng  http://orcid.org/0000-0002-4565-8717

## FUNDING

| Funder | Grant(s) | Author(s) |
| --- | --- | --- |
| National Natural Science Foundation of China | 42276093 | Yuanyuan Feng |
| National Basic Research Program of China | 2023YFC3108400 | Dahai Zhang |
| Deep Blue Foundation of Shanghai Jiao Tong University | SL2022PT203 | Yuanyuan Feng |

## AUTHOR CONTRIBUTIONS

Jiahui Zheng, Data curation, Investigation, Methodology, Writing – original draft, Writing – review and editing | Jumao Yuan, Investigation, Methodology | Xinwei Wang, Conceptualization, Formal analysis, Methodology | Dahai Zhang, Conceptualization, Methodology, Supervision | Lin Lin, Conceptualization, Methodology, Supervision | Nuo Shi, Conceptualization, Formal analysis, Funding acquisition, Supervision | Yuanyuan Feng, Conceptualization, Funding acquisition, Investigation, Methodology, Project administration, Resources, Supervision, Writing – original draft, Writing – review and editing

## DATA AVAILABILITY

The research data are available at https://data.mendeley.com/datasets/fnbh4sxm9w/1.

## ADDITIONAL FILES

The following material is available online.

## Supplemental Material

**Supplemental material (Spectrum03449-25-s0001.docx).** Fig. S1 and S2, Tables S1 to S3.

## Open Peer Review

**PEER REVIEW HISTORY (review-history.pdf).** An accounting of the reviewer comments and feedback.

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
