## [Reviewer comments · Microbiology Spectrum]

Microbiology Spectrum

Intermittent light enhances pigment production in the diatom *Phaeodactylum tricornutum*: A combined physiological and transcriptomic approach

Jiahui Zheng, Jumao Yuan, Xin-Wei WANG, Dahai Zhang, Lin Lin, Nuo Shi, and Yuanyuan Feng

Corresponding Author(s): Yuanyuan Feng, Shanghai Jiao Tong University

Review Timeline:

Submission Date:	October 23, 2025
Editorial Decision:	December 8, 2025
Revision Received:	January 14, 2026
Accepted:	March 12, 2026

Editor: Adriana Lopes dos Santos

Reviewer(s): Disclosure of reviewer identity is with reference to reviewer comments included in decision letter(s). The following individuals involved in review of your submission have agreed to reveal their identity: Xin Lin (Reviewer #2)

Transaction Report:

DOI: <https://doi.org/10.1128/spectrum.03449-25>

Re: Spectrum03449-25 (**Intermittent light enhances pigment production in the diatom *Phaeodactylum tricornutum*: A combined physiological and transcriptomic approach**)

Dear Dr. Yuanyuan Feng:

Thank you for the privilege of reviewing your work. Below you will find my comments, instructions from the Spectrum editorial office, and the reviewer comments.

Revision Guidelines

Sincerely,
Adriana Lopes dos Santos
Editor
Microbiology Spectrum

Reviewer #1 (Comments for the Author):

This study investigated the physiological and molecular adaptation mechanisms of the model diatom *Phaeodactylum tricornutum* under simulated seasonal and intermittent light-dark (LD) cycles. The authors found that intermittent light enhanced growth and photosynthetic efficiency, with the highest pigment accumulation observed under the LD 2:2 regime. This regime also increased the ratio of photosynthetic to photoprotective pigments and reduced the de-epoxidation of diadinoxanthin. Transcriptomic

analysis revealed that moderate LD cycles (LD 2:2 and LD 12:12) upregulated genes involved in chlorophyll and carotenoid biosynthesis as well as those related to light-harvesting complex pathways.

This manuscript presents a well-designed and timely investigation of the physiological and molecular responses of *Phaeodactylum tricornutum* to fluctuating light-dark regimes. The integration of physiological measurements with transcriptomic analysis provides a comprehensive understanding of how diatoms modulate pigment biosynthesis and photosynthetic processes under dynamic light environments. Overall, the manuscript is well structured and the results are clearly presented, offering valuable insights into diatom acclimation strategies, with potential implications for both ecological studies and industrial pigment production.

The strength of the work is the systematic comparison across multiple LD cycles, including intermittent LD cycles, which allows the identification of optimal light regimes (e.g., LD 2:2) that enhance growth and pigment yield. However, several issues remain to be addressed, and the manuscript should be revised accordingly.

1. Materials and Methods:

In the section of "Experimental Design and Growth Conditions", it is mentioned that "the cultures were acclimated to the respective light regimes for 14 days under semi-continuous incubation, with daily dilution using freshly made medium. After acclimation, cultures in the exponential growth phase were inoculated into 2 L polycarbonate bottles.." Please clarify the cell abundance and the volume of incubation bottles used for the semi-continuous incubation period - this would determine whether the light regimes were comparable to the final incubation period.

2. The full Latin name *Phaeodactylum tricornutum* and its abbreviated form *P. tricornutum* are used inconsistently throughout the manuscript, which is not in accordance with standard scientific writing conventions. I suggest using the full name *Phaeodactylum tricornutum* only at its first appearance, and the abbreviated form *P. tricornutum* thereafter.

3. In the Results section, for the description of cellular pigment contents and elemental contents, I suggest that these values be rounded to two decimal places for clarity.

4. The study could be further strengthened by expanding the discussion on the ecological relevance of intermittent light regimes in natural oceanic conditions.

Reviewer #2 (Comments for the Author):

Diatoms are crucial primary producers in the ocean. Their pigments contribute significantly to their ecological success and hold promise for biotechnological applications. Although most molecular synthesis pathways of diatom pigments have been well studied, the physiological and molecular responses to intermittent light environments are worthy of investigation. The authors investigated the physiological and transcriptomic response under different light and dark regimes. The most interesting result in this study is that short intervals of light and darkness can boost diatom growth and increase pigment production, especially under very short light cycles.

This study demonstrates that diatoms were able to adapt to variable light environments, which has important ecological implications and highlights their potential for improved pigment production in industry.

The manuscript is already well-structured and clearly written. To make it even stronger, we recommend focusing on further refining the clarity in a few sections and more explicitly articulating the scientific significance of the work.

1. In the abstract, the authors mentioned "simulated seasonal and intermittent light-dark (LD) cycles". However, in the manuscript no words on the seasonal light dark cycles were explained. LD cycles are correlated with season dynamics. I suggest authors add one or two sentence to explain this.

2.2.5 Transcriptomic analysis

"Culture cells from the late exponential growth phase (100 ML)." Could you specify the sample collection time points for different treatments? In the light or dark cycle ?

3. The specific growth rate of LD(2:4) was significantly lower than other treatments. Any explanation ?

4. Fig 2 A , the y axis should be "pigment content ug/L"

5. The RNA-seq data (specifically, the FPKM values for all genes across the different treatments) could be provided as an Excel spreadsheet to serve as a resource for researchers interested in this dataset and to promote future studies.

6. The ecological significance of this study should be mentioned briefly in the discussion.

7. The light intensity you used was $70 \mu\text{mol photons}\cdot\text{m}^{-2}\cdot\text{s}^{-1}$. The rationale for the specific light intensity used in this study should be clearly stated. Furthermore, we recommend briefly discussing the potential interaction between the chosen light intensity and the different light-dark regimes, as this would be valuable for interpreting the results.

Diatoms are crucial primary producers in the ocean. Their pigments contribute significantly to their ecological success and hold promise for biotechnological applications. Although most molecular synthesis pathways of diatom pigments have been well studied, the physiological and molecular responses to intermittent light environments are worthy of investigation. The authors investigated the physiological and transcriptomic response under different light and dark regimes. The most interesting result in this study is that short intervals of light and darkness can boost diatom growth and increase pigment production, especially under very short light cycles.

This study demonstrates that diatoms were able to adapt to variable light environments, which has important ecological implications and highlights their potential for improved pigment production in industry.

The manuscript is already well-structured and clearly written. To make it even stronger, we recommend focusing on further refining the clarity in a few sections and more explicitly articulating the scientific significance of the work.

1. In the abstract, the authors mentioned “simulated seasonal and intermittent light–dark (LD) cycles” . However, in the manuscript no words on the seasonal light dark cycles were explained. LD cycles

are correlated with season dynamics. I suggest authors add one or two sentence to explain this.

2. 2.5 Transcriptomic analysis

“Culture cells from the late exponential growth phase (100 ML).”

Could you specify the sample collection time points for different treatments? In the light or dark cycle?

3. The specific growth rate of LD(2:4) was significantly lower than other treatments. Any explanation?

4. Fig 2 A, the y axis should be “pigment content ug/L”

5. The RNA-seq data (specifically, the FPKM values for all genes across the different treatments) could be provided as an Excel spreadsheet to serve as a resource for researchers interested in this dataset and to promote future studies.

6. The ecological significance of this study should be mentioned briefly in the discussion.

7. The light intensity you used was $70 \mu\text{mol photons}\cdot\text{m}^{-2}\cdot\text{s}^{-1}$. The rationale for the specific light intensity used in this study should be clearly stated. Furthermore, we recommend briefly discussing the potential interaction between the chosen light intensity and the different light-dark regimes, as this would be valuable for interpreting the results.

Dear editor and reviewers,

We really appreciate your helpful comments and suggestions on the manuscript, which have greatly helped to improve the overall quality of the manuscript. Now we have thoroughly revised the m/s according to your comments and uploaded the revised version in the system.

Our point-by-point responses to the issues that you raised are listed as below.

Reviewer #1

1. Materials and Methods:

In the section of "Experimental Design and Growth Conditions", it is mentioned that "the cultures were acclimated to the respective light regimes for 14 days under semi-continuous incubation, with daily dilution using freshly made medium. After acclimation, cultures in the exponential growth phase were inoculated into 2 L polycarbonate bottles.." Please clarify the cell abundance and the volume of incubation bottles used for the semi-continuous incubation period - this would determine whether the light regimes were comparable to the final incubation period.

Response:

The volume of the incubation bottles for the stock culture was 1 L and the average cell abundance was maintained $\sim 2 \times 10^6$ cells/mL, comparable to that during the final incubation period. As such, the light conditions during the two growth phases were also similar.

These details have been added in lines 137-140: "Prior to the start of the batch incubation, the cultures were acclimated to the respective light regimes in 1 L incubation bottles for 14 days under semi-continuous incubation, with daily dilution using freshly made medium to maintain the cell abundance $\sim 2 \times 10^6$ cells/mL."

2. The full Latin name *Phaeodactylum tricornutum* and its abbreviated form *P. tricornutum* are used inconsistently throughout the manuscript, which is not in accordance with standard scientific writing conventions. I suggest using the full name

Phaeodactylum tricornutum only at its first appearance, and the abbreviated form *P. tricornutum* thereafter.

Response:

The abbreviated form *P. tricornutum* has been revised consistently throughout the manuscript.

3. In the Results section, for the description of cellular pigment contents and elemental contents, I suggest that these values be rounded to two decimal places for clarity.

Response:

These values have been rounded to two decimal places in the revised version.

4. The study could be further strengthened by expanding the discussion on the ecological relevance of intermittent light regimes in natural oceanic conditions.

Response:

The ecological relevance of intermittent light regimes in natural oceanic conditions has been added in the discussion section in 892-895:” Our results reveal that higher frequency of light fluctuation may promote both photosynthetic performance and pigment synthesis efficiency. This suggests a potential acclimation strategy for diatoms in dynamic marine environments, such as turbulent waters with fast mixing or regions subject to periodic cloud cover”.

Reviewer #2

1. In the abstract, the authors mentioned "simulated seasonal and intermittent light-dark (LD) cycles". However, in the manuscript no words on the seasonal light dark cycles were explained. LD cycles are correlated with season dynamics. I suggest authors add one or two sentence to explain this.

Response:

In the introduction, the following sentences have been added in lines 89-91: “In addition, another important environmental factor regulating the growth and photosynthesis of phytoplankton is the periodical fluctuation of light and dark condition, including both diel and short-term variations”, and lines 114-117: “In order to further understand the effects of photoperiods on different growth stages in marine diatoms and reveal the underlying molecular regulatory mechanisms, we carried out laboratory incubation experiments on diatom *P. tricornutum* under seven distinct light-dark cycles to simulate a range of seasonal and intermittent light-dark fluctuations”.

In addition, in the discussion section, the ecological relevance of intermittent light regimes in natural oceanic conditions has been added in lines 892-895:” Our results reveal that higher frequency of light fluctuation may promote both photosynthetic

performance and pigment synthesis efficiency. This suggests a potential acclimation strategy for diatoms in dynamic marine environments, such as turbulent waters with fast mixing or regions subject to periodic cloud cover”.

2. 2.5 Transcriptomic analysis

"Culture cells from the late exponential growth phase (100 ML)." Could you specify the sample collection time points for different treatments? In the light or dark cycle?

Response:

The samples were all collected in the light cycle. This has been also specified in the revised m/s in line 226: "Culture cells (100 mL) from the late exponential growth phase at the light phase were filtered onto 0.6 µm polycarbonate membranes (Millipore, USA)".

3. The specific growth rate of LD(2:4) was significantly lower than other treatments. Any explanation?

Response: The low specific growth rates at LD 2:4 and LD 8:16 were probably caused by the shorter light durations, as indicated in lines 603-604: "shorter light durations (LD 8:16, LD 2:4) restricted total light energy acquisition, resulting in reduced growth rates."

In addition, although the average growth rate at LD 2:4 was lower than other treatments, the statistical analysis suggested no significant difference between LD 2:4 and 8:16 were not statistically different, shown in Fig. 1b.

4. Fig 2 A, the y axis should be "pigment content ug/L"

Response:

This has been corrected in the revised figure 2.

5. The RNA-seq data (specifically, the FPKM values for all genes across the different treatments) could be provided as an Excel spreadsheet to serve as a resource for researchers interested in this dataset and to promote future studies.

Response:

These values now have been provided in an excel file in the supplementary materials.

In addition, the data file is now available online at:

<https://data.mendeley.com/datasets/fnbh4sxm9w/1>

6. The ecological significance of this study should be mentioned briefly in the discussion.

Response:

The ecological significance of this study, especially the ecological relevance of intermittent light regimes in natural oceanic conditions has been added in the

discussion section in lines 892 - 895: "Our results reveal that higher frequency of light fluctuation may promote both photosynthetic performance and pigment synthesis efficiency. This suggests a potential acclimation strategy for diatoms in dynamic marine environments, such as turbulent waters with fast mixing or regions subject to periodic cloud cover".

7. The light intensity you used was $70 \mu\text{mol photons}\cdot\text{m}^{-2}\cdot\text{s}^{-1}$. The rationale for the specific light intensity used in this study should be clearly stated. Furthermore, we recommend briefly discussing the potential interaction between the chosen light intensity and the different light-dark regimes, as this would be valuable for interpreting the results.

Response:

The light intensity ($70 \mu\text{mol photons}\cdot\text{m}^{-2}\cdot\text{s}^{-1}$) used in the study was close to the commonly used optimal light intensity for fucoxanthin production (Gomez-Loredo et al., 2016; Dong et al., 2023). This has also been added in the revised m/s in lines 164-165: "The light intensity used for the incubation was close to the optimal condition for fucoxanthin production of *P. tricorutum* ^[31,32]."

Thank you for your efforts in handling the manuscript. We look forward to hearing back from you again.

Sincerely,

Yuanyuan Feng and Jiahui Zheng

Re: Spectrum03449-25R1 (**Intermittent light enhances pigment production in the diatom *Phaeodactylum tricornutum*: A combined physiological and transcriptomic approach**)

Dear Dr. Yuanyuan Feng:

Your manuscript has been accepted, and I am forwarding it to the ASM production staff for publication. Your paper will first be checked to make sure all elements meet the technical requirements. ASM staff will contact you if anything needs to be revised before copyediting and production can begin. Otherwise, you will be notified when your proofs are ready to be viewed.

Sincerely,
Adriana Lopes dos Santos
Editor
Microbiology Spectrum